# Effect of Ultra-Fine Cement on the Strength and Microstructure of Humic Acid Containing Cemented Soil

**Jing Cao** [1], **Fangyi Liu** [1,*], **Zhigang Song** [1], **Wenyun Ding** [2], **Yongfa Guo** [2], **Jianyun Li** [1] **and Guoshou Liu** [1]

[1] Faculty of Civil Engineering and Mechanics, Kunming University of Science and Technology, Kunming 650500, China; 20040130@kust.edu.cn (J.C.)

[2] Kunming Survey, Design and Research Institute Co., Ltd. of CREEC, Kunming 650200, China

[*] Correspondence: 20202210062@stu.kust.edu.cn

**Abstract:** The peat soil in the Dianchi Lake area of Yunnan, China, is widely distributed, bringing many problems to engineering. The peat soil foundation is usually treated by the cement mixing method, and the reinforcement effect of cemented soil is mainly affected by humic acid (HA). Ultra-fine cement (UFC) can improve cement performance and reduce cement consumption, decreasing $CO_2$ emissions and the impact of human activities on the environment. Simulated peat soils in different environments are prepared with HA reagent and cohesive soil, reinforced by composite cement curing agent mixed with ultrafine cement (UFC). The relationship among the UFC proportion, HA reagent content, soaking time, and sample strength was studied. The unconfined compressive strength test (UCS), scanning electron microscope (SEM), and PCAS microscopic quantitative test techniques were used to explore the mechanism of the effect of UFC on the strength of HA-containing cemented soil. The increasing UFC proportion in the composite cement curing agent gradually increased HA-containing cemented soil's strength. UFC significantly reduced the percentage of macropores in HA-containing cemented soil and made the microstructure denser. The HA-containing cemented soil's $q_u$ increased the most when the UFC proportion increased from 0% to 10%. The solidification effect of the composite cement curing agent mixed with UFC was always stronger than that of OPC. The composite cement curing agent with a UFC proportion of 10% is practical. Cement is still an important building material in the current construction industry, and UFC provides a new method for reducing environmental impact in engineering construction.

**Keywords:** peat soil; humic acid; cemented soil; ultra-fine cement; UCS; SEM; PCAS

## 1. Introduction

Cement is an important building material in the construction industry and is widely used in various fields of national economic construction. China produces the most cement worldwide, but the carbon emissions generated by the cement industry cannot be underestimated [1,2]. Cement production emits high $CO_2$ [3–5]. Although China's cement industry's carbon emissions account for a relatively low proportion of the country's carbon emissions, it accounts for more than half of industrial process emissions [6]. In addition, $SO_2$ and $NO_x$ emitted by the cement industry will cause environmental disasters, such as smog and acid rain. China has listed the cement industry as one of the key industries for $SO_2$ and $NO_x$ emissions reduction during the "14th Five-Year Plan" period [7]. To achieve China's carbon peaking and carbon neutrality goals and to improve the natural environment, reducing the cement consumption is an urgent problem to be solved.

Improving cement performance can meet the design and construction requirements of actual engineering with less cement consumption. The performance of cement is affected by many factors. Considering the physical properties of cement, the particle size distribution and specific surface area of cement will significantly impact the cement's performance [8–10]. On the premise of not changing the chemical composition and relative

content of cement, the physical properties of cement particles can be improved to enhance its performance and reduce the cement consumption used in actual engineering. The reduction in cement consumption reduces the impact of human activities on the environment based on the collection, production, and use of cement raw materials. There is no uniform standard for the definition of UFC at home and abroad, so this study generally assumed that ultrafine cement needs to meet the requirements of particle size or specific surface area (maximum particle size < 20 μm or specific surface area > 8000 cm$^2$/g) [11–14]. The particles of UFC are smaller than OPC particles and have a greater specific surface area and chemical activity. Soft soil foundations are usually reinforced by cement deep mixing, but there have been few related studies on cemented soil to strengthen peat soil. Humic acid (HA) is an important component of humus in peat soil, and its impact on cemented soil is higher than other components [15,16]. Strength is the core index of cemented soil design and construction control, and the study of the effect of UFC on the strength of HA-containing cemented soil has guiding significance for cemented soil use to strengthen peat soil foundations.

In 1993, Reinhardt et al. [12] found through research on UFC that it has finer particles than OPC, has better compatibility than other chemical caulking materials and is suitable for filling concrete cracks. In 1999, Hidenori et al. [17] found through experimental research that the permeability of UFC slurry is affected by particle distribution. In 2008, Celik et al. [18] studied the influence of cement particle size distribution, distribution uniformity, and specific surface area on the performance of cement. Their tests showed that the fineness of cement will affect the hydration reaction and the development of early strength. In 2018, Wu et al. [19] developed a new type of green cement using ultra-fine cement and high-volume solid waste (fly ash and blast furnace slag) as raw materials to reduce carbon emissions and environmental impact. The mechanical properties of the new green cement are significantly improved compared with the traditional ordinary Portland cement. In 2021, Mollamahmutoglu et al. [20] found, through grouting performance tests on UFC containing boric acid additives, that adding boric acid additives improved the groutability of UFC grouting and reduced its strength and permeability. Consequently, the research on UFC at home and abroad has mostly concentrated on the construction of grouting engineering. At the same time, there has been less research on UFC as a curing agent to strengthen soft soil foundations.

Regarding the engineering characteristics of peat soil, relevant scholars have proved that HA will affect the cemented soil strength, but a practical solution has yet to be proposed. The production cost of UFC is relatively high, so it is outside the scope of engineering practice to use it entirely in foundation treatment. In this paper, HA is added to cohesive soil to simulate peat soil, and different UFC proportions are added to OPC to form a composite cement curing agent to solidify peat soil. Based on previous studies, the influence of UFC on the strength development of HA-containing cemented soil has been revealed, and the possibility of reducing the cement consumption in actual engineering has been explored. The relevant mechanism has been further analyzed using scanning electron microscopy (SEM) and PCAS microscopic quantitative testing techniques.

The cement mixing method is commonly used to deal with soft soil foundations. The research results will provide a theoretical basis for UFC to treat peat soil foundations and ideas for using UFC to treat different types of soft soil foundations. The application of UFC will provide new solutions for reducing carbon emissions and reducing environmental impact in engineering construction.

## 2. Materials and Methods

### 2.1. Experimental Materials

Figure 1 is the test soil collection location. The test soil was taken from a soil slope of Chenggong District, Kunming City, and its organic matter content was found to be extremely low after humic acid testing. The test soil was collected at a distance of about

15 cm from the ground surface. The test soil was ground after exposure to the sun, and the soil material sieved from the 2.00-mm sieve was sealed and stored.

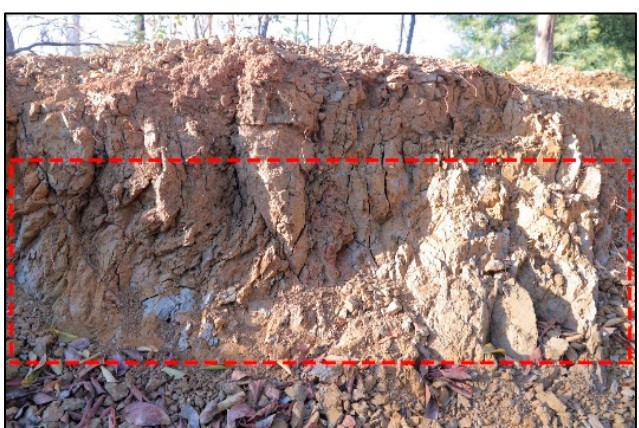

**Figure 1.** Collection of test soil. The red dotted line box marks the location of collecting test soil.

Table 1 shows the relevant indicators of the test soil measured by the "Standard for geotechnical testing method" (GB/T 50123-2019) [21].

**Table 1.** Physical mechanical index of testing soil.

| Test Soil | Natural Moisture Content (%) | Plastic Limit (%) | Liquid Limit (%) | Natural Density (g/cm$^3$) | Specific Gravity of Soil Particle (Gs) |
|---|---|---|---|---|---|
| Cohesive soil | 18.6 | 23.0 | 39.2 | 1.96 | 2.73 |

Figure 2 is the X-ray powder diffraction pattern of test soil, using a Holland PANalytical X'Pert3 Powder type multifunctional powder X-ray diffractometer to obtain it. The diffraction peaks of the test soil were analyzed using Jade software, version 6.0. The test soil mainly comprised quartz ($SiO_2$) and kaolinite ($Al_2O_3 \cdot 2SiO_2 \cdot 2H_2O$), and its mineral properties were relatively stable.

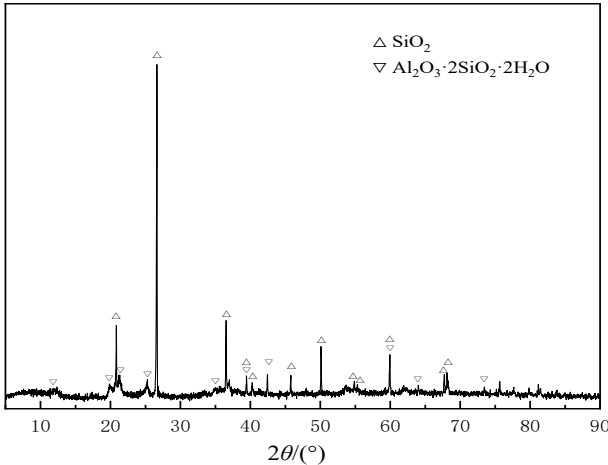

**Figure 2.** XRD pattern of test soil.

Figure 3 is a cumulative particle size distribution curve obtained using a Malvern Mastersizer 2000 laser particle size analyzer on the test materials. The maximum particle size of the test soil was much larger than that of the other test materials. According to the test soil's particle size and plasticity index, it met the definition of cohesive soil in the

"Code for design of building foundation" (GB 50007-2011) [22]. The maximum particle size of the HA reagent is second only to cohesive soil, and the content of particles larger than 10 μm was about 64.1%. If the relevant chemical reactions are not considered, when the humic acid reagent is mixed into the cohesive soil, the two constitute the skeleton of the simulated peat soil. The actual HA content in the HA reagent is calculated based on the HA carbon content range to be about 41.68% [23,24]. Ordinary Portland cement (OPC) is a P·O 42.5 type Portland cement produced according to China's "Common Portland cement"(GB 175-2007) [25]. The two materials' particle sizes can be significantly different from the cumulative curves of OPC and UFC. The maximum particle size of UFC is 11.8 μm, which is much smaller than that of OPC, which is 138.0 μm. UFC is prepared from OPC through physical grinding (ball mill).

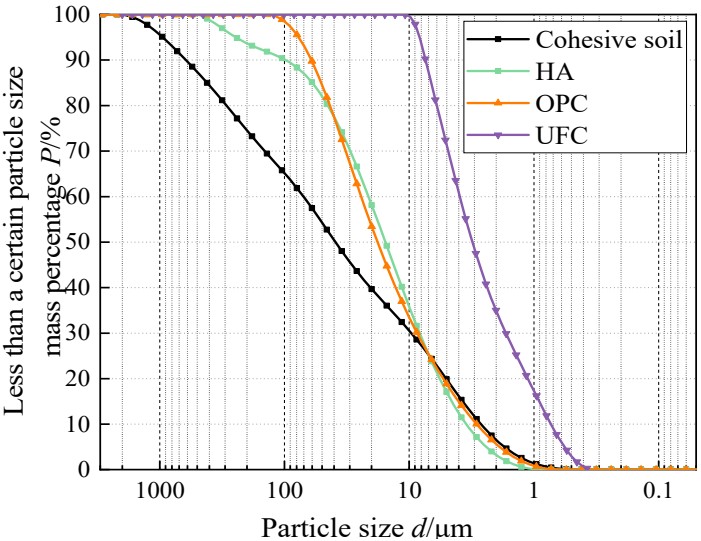

**Figure 3.** Cumulative particle size curve of test materials.

### 2.2. Experimental Method

Considering the practicability and economy of the actual engineering, different UFC proportions $\gamma$ (0%, 10%, 20%, 30%, 40%, 50%) were set with a gradient of 10%. The HA content of peat soil in the Dianchi Lake area is about 2.36% to 28.13% [26]. The HA reagent content $\lambda$ (0%, 15%, 30%) was set according to the actual HA content, and the HA reagent was mixed with cohesive soil to simulate different peat soil environments. At the same time, different soaking times (28 d, 90 d, 180 d, 270 d, 365 d) were set to study the strength development law of samples with different UFC proportions. The specific test plan is shown in Table 2.

**Table 2.** Test scheme.

| Tests | Cement Rate $\beta$/% | Proportion of UFC $\gamma$/% | HA Reagent Content $\lambda$/% | Soaking Time/d |
|:---:|:---:|:---:|:---:|:---:|
| UCS | 20 | 0, 10, 20, 30, 40, 50 | 0, 15, 30 | 28, 90, 180, 270, 365 |
| SEM, PCAS | | | 15 | 90 |

The water content ($w$ = 24%) and void ratio ($e$ = 0.8) of the sample were set in the test, and the mass of the test material was calculated according to Formulas (1)–(3). The samples were made according to the manufacturing process of the unconfined compressive strength samples in the "Standard for geotechnical testing method" (GB/T 50123-2019) [21]. The test material was placed in a blender and stirred evenly, and then distilled water was added to continue stirring. After the test materials were stirred evenly, we put them into the mold (inner diameter d = 39.1 mm, height h = 80 mm) in turn to make standard samples.

Disintegration would occur if the sample were directly immersed in distilled water. The samples needed to be cured to a certain strength before soaking. The production process of the sample is shown in Figure 4.

$$\beta = \frac{m_{s(OPC)} + m_{s(UFC)}}{m_{s(HA)} + m_{s(soil)}} \times 100\% \tag{1}$$

$$\gamma = \frac{m_{s(UFC)}}{m_{s(OPC)} + m_{s(UFC)}} \times 100\% \tag{2}$$

$$\lambda = \frac{m_{s(HA)}}{m_{s(HA)} + m_{s(soi)}} \times 100\% \tag{3}$$

where $m_{s(OPC)}$ is the OPC particle weight, $m_{s(UFC)}$ is the UFC particle weight, $m_{s(HA)}$ is the HA particle weight, and $m_{s(soil)}$ is the cohesive soil particle weight.

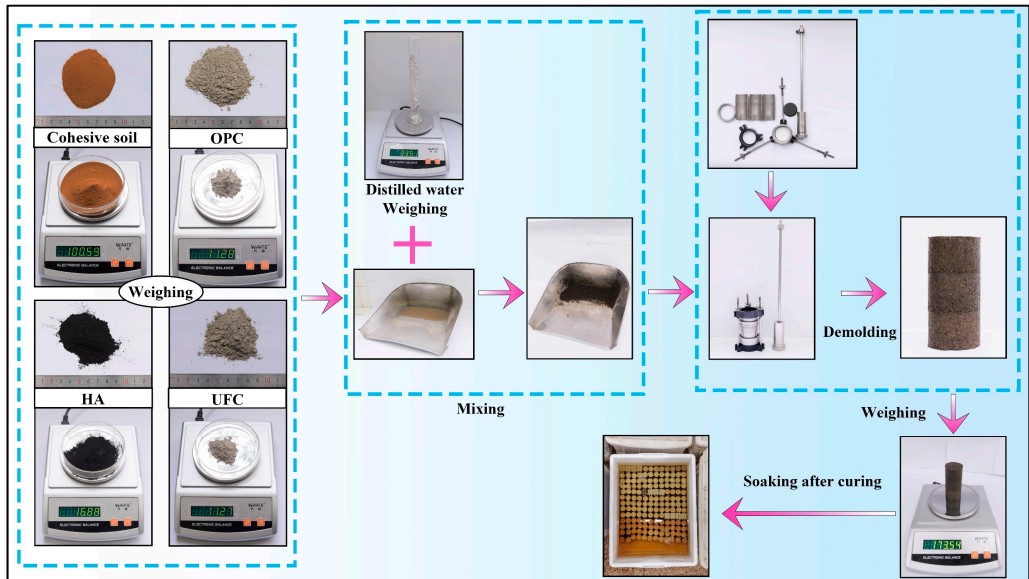

**Figure 4.** Sample production process.

### 2.3. Experimental Program

The strength test used a conventional lime soil unconfined compression tester, and the lifting speed of the tester was 1 mm/min. The unconfined compressive strength test (UCS) complied with the relevant provisions of the "Standard for geotechnical testing method" (GB/T 50123-2019) [21].

After the sample was dried at 50 °C, it was made into a block suitable for the sample stage. Block samples needed to be sprayed with gold. Appropriate block samples were placed in a Czech TESCAN-VEGA3 electron microscope for scanning electron microscopy (SEM). Particles (Pores) and Cracks Analysis System (PCAS) software (version V2.3) was used to divide and count the pores of the SEM images.

## 3. UCS Results and Analysis

### 3.1. Influence of HA on Cemented Soil

Figure 5 shows the relationship between the unconfined compressive strength ($q_u$) and humic acid (HA) reagent content $\lambda$ for cemented soil samples with different soaking times (the soaking times were 28 d, 90 d, 180 d, 270 d, and 365 d). The cemented soil $q_u$ decreased gradually with the increase in HA reagent addition. The samples $q_u$, the soaking time of which was less than 270 d, increased gradually with soaking time. The samples with soaking times of 28 d, 90 d, 180 d, and 270 d were in the stage of strength development.

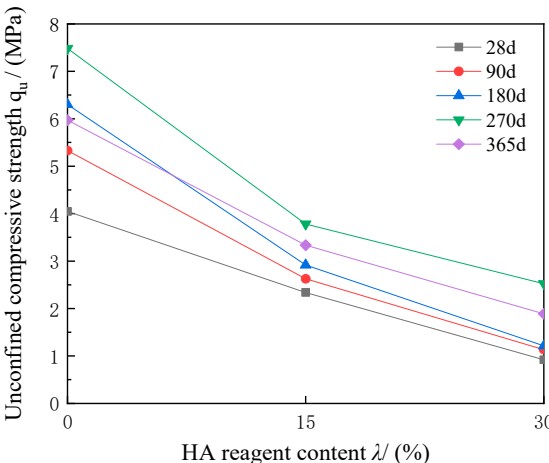

**Figure 5.** The relationship curve between $q_u$ and the HA reagent content of cemented soil with different soaking times.

The reduction in cemented soil strength caused by HA is mainly due to the following aspects. The increase in HA reagent content in cement soil causes humic acid particles to bear more stress in the skeleton. HA particles have higher compressibility than other inorganic minerals in the soil, making cemented soil's strength and compressive capacity with HA as the skeleton weaker [27]. HA particles are finer and negatively charged compared to cohesive soil particles. The surface of cement particles and soil particles will adsorb HA particles, affecting the hydration reaction [28]. Adding HA particles to cohesive soil disperses the cohesive soil particles and enhances the fluidity, which is not conducive to the curing of cement [29].

### 3.2. Influence of UFC on Cemented Soil

3.2.1. Influence of UFC Proportion on Cemented Soil

Figure 6 shows the relationship between the unconfined compressive strength ($q_u$) of the sample and the UFC proportion in the composite cement curing agent. The figure also compares the samples $q_u$ with different humic acid reagent content under each UFC proportion $\gamma$. By comparing the strength data of samples with different soaking times, it is found that the strength of cemented soil with different HA content increased with UFC proportion.

Scrivener, Tsakalakis, Celik, et al.'s [30–33] studies have shown that cement particles with a particle size less than 32 μm have a fast hydration rate and improved early strength. The cement particle size is 32 μm–60 μm, and the hydration reaction speed is relatively slow, mainly providing the later strength. Cement particles larger than 60 μm are mainly coarse particles, less involved in hydration reactions, and they make limited contributions to strength. The percentages of ordinary Portland cement (OPC) and UFC particles smaller than 32 μm used in the test were 67.68% and 100%, respectively. The fine particles of the composite cement reinforcement composed of OPC and UFC gradually increased with the UFC proportion, and the mass percentage of cement particles smaller than 32 μm gradually increased. The hydration products increase continuously with the progress of the hydration reaction, and the hydration products gradually fill the through pores to become closed pores. The structure of cemented soil is enhanced as the pores are gradually filled. The reinforcement effect of the composite cement curing agent on the simulated peat soil increases with UFC proportion.

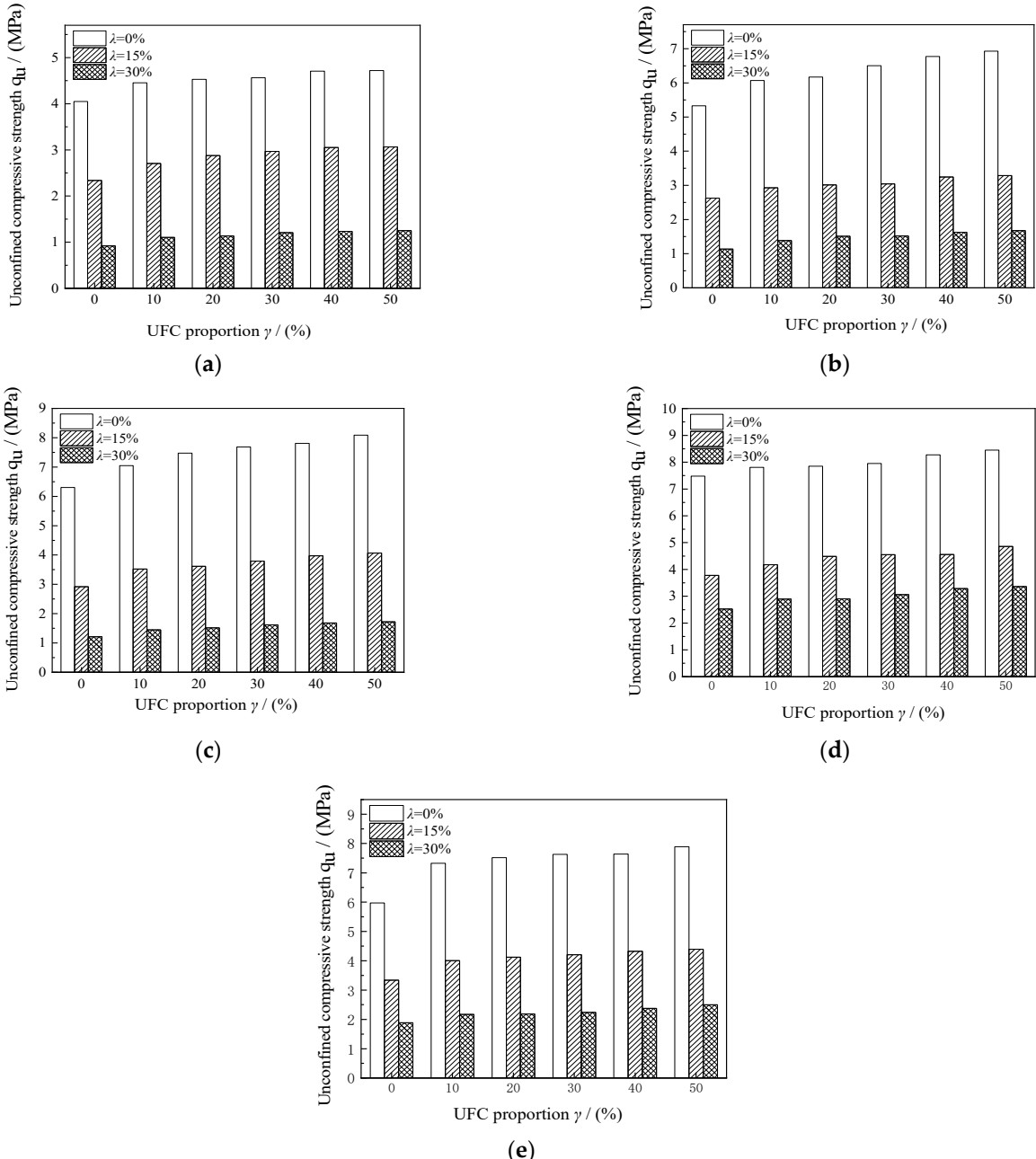

**Figure 6.** The histogram of the relationship between $q_u$ and the UFC proportion of cemented soil with different HA reagent content $\lambda$. (**a**) Soaking time is 28 d. (**b**) Soaking time is 90 d. (**c**) Soaking time is 180 d. (**d**) Soaking time is 270 d. (**e**) Soaking time is 365 d.

The cemented soil strength is affected by porosity and pore size distribution [34,35]. The speed of generating hydration products is weakened with a UFC proportion of more than 10%. Calcium silicate hydrate with cementation is a hydration product that contributes to cemented soil strength, and its thermodynamic equilibrium formula is as follows (4) [36,37]. As the pores are gradually closed, the concentration of $Ca^{2+}$ and $OH^-$ in the closed pore solution gradually decreases, reducing the formation rate of calcium silicate hydrate. Therefore, the relationship between the sample $q_u$ and the UFC proportion did not increase linearly. The strength growth rate of the sample slowed when the UFC proportion was more than 10%.

$$6Ca^{2+}(aq.) + 5HSiO_3^-(aq.) + 7OH^-(aq.) \rightleftharpoons 6CaO \cdot 5SiO_2 \cdot 6H_2O(C\text{-}S\text{-}H) \tag{4}$$

### 3.2.2. Influence of Soaking Time on Cemented Soil

Figure 7 shows the relationship between the unconfined compressive strength ($q_u$) of the sample and the soaking time. The picture also compares the strength of the samples under the composite cement curing agent with different UFC proportions $\gamma$. In the stage of strength development (soaking times of 28 d, 90 d, 180 d, and 270 d), the samples' $q_u$ increased with the soaking time. The samples $q_u$ showed different degrees of decline after soaking for more than 270 d. The sample $q_u$ with 30% HA reagent incorporation increased significantly in the later stage of strength development (the soaking time increased from 180 d to 270 d). The addition of UFC increased the cemented soil $q_u$.

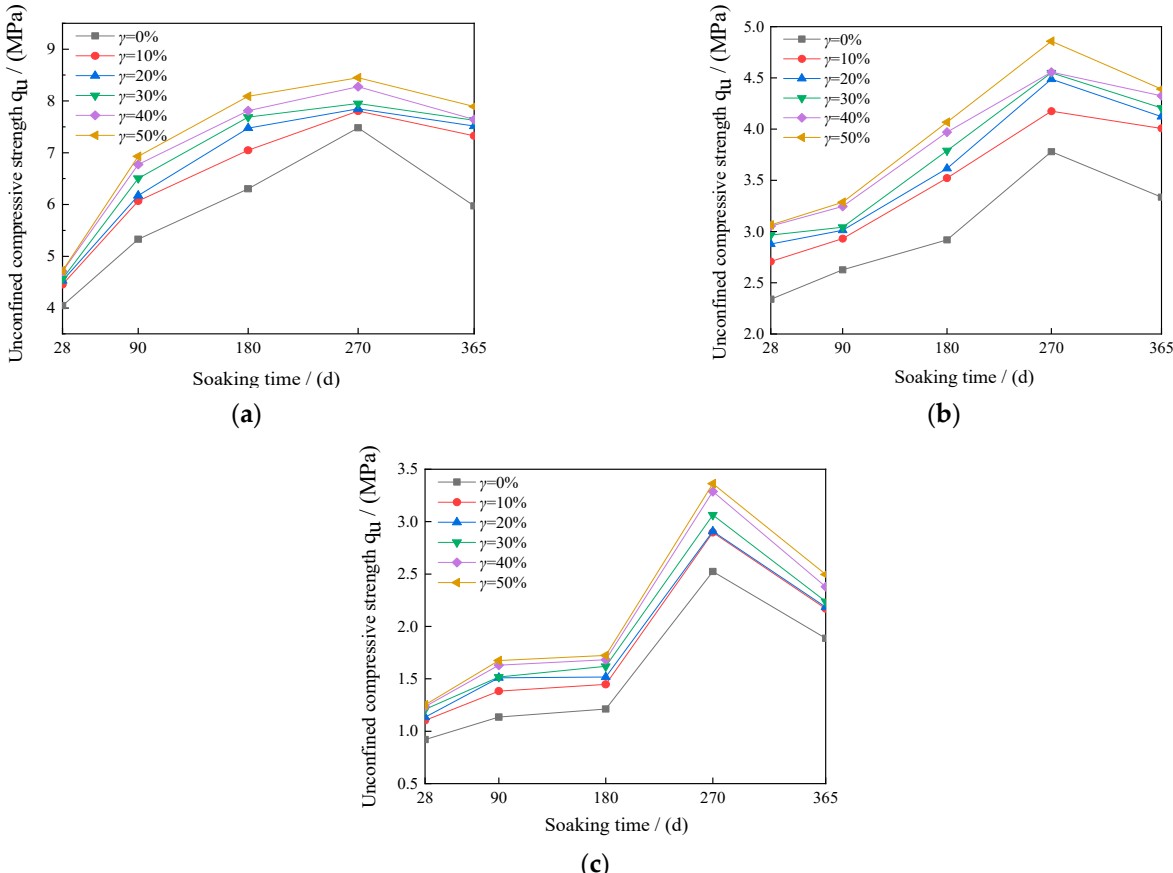

**Figure 7.** The curve of the relationship between $q_u$ and soaking time of samples. (**a**) HA reagent content $\lambda$ = 0%. (**b**) HA reagent content $\lambda$ = 15%. (**c**) HA reagent content $\lambda$ = 30%.

The hydration reaction proceeds continuously during the strength development stage of the cemented soil, and the environment in which the sample is located remains alkaline. The alkaline environment further promotes the reaction of cement particles and clay minerals with an alkali solution, accelerates the hydration reaction, and improves the microscopic structure of the sample [38]. An alkaline environment with high pH will affect the $q_u$ development of HA-containing cemented soil, and the cemented soil strength will decrease [39,40]. The hydration reaction in cemented soil is basically completed at around 270 d, and cemented soil's disintegration begins to disintegrate. The metal cations inside and outside the sample make the double electron layer between soil particles thicker, and the repulsive force between particles is greater than the attractive force, causing the internal structure of cemented soil to tend to be destroyed [41]. The aqueous solution continuously infiltrates the internal pores of the cemented soil sample with soaking, promoting the hydration reaction and gradually sealing and compressing a small amount of air in the pores to form a "micro-air chamber" [42]. The existence of the "micro-air chamber" causes the sample's internal structure to bear a certain pressure, affecting the connection between the

particles in the sample. The pH value of the water tank is kept in the range of 11–12 before sampling (soaking times are 270 d and 365 d, respectively). Under alkaline corrosion and disintegration, the cemented soil forms cracks and pores, leading to a 365-d $q_u$ decrease.

HA has the property of being slightly soluble in alkaline solutions [16]. The hydration reaction will put the sample in an alkaline environment, and the HA particles will gradually dissolve. The dissolution of HA makes the structure of cemented soil loose [42]. HA also readily forms calcium humate with $Ca^{2+}$ [43–46]. The formation of HA reduces the formation of hydration products (combined with Formula (4)). The hydration reaction is affected by humic acid to produce fewer hydration products, and the partial dissolution of HA particles leads to the loosening of the microscopic structure of the cemented soil. Both of them affect the strength development of the cemented soil. Compared with Figure 7b,c, it can be seen that there is a soaking time threshold for the inhibitory effect of HA on cemented soil in an environment with a high HA reagent content. When the soaking time increased from 180 d to 270 d, the cemented soil $q_u$ with 30% HA reagent content increased greatly, and the reinforcement effect of the hydration reaction was stronger than the adverse effect of HA on cemented soil. There is also a threshold for the possible interaction of HA with hydration reactions. In a low HA environment (HA reagent content is 15%), HA has weak inhibition on hydration, and the interaction between the hydration reaction and HA is near the threshold value, causing the $q_u$ (soaking time of HA-containing cemented soil to increase from 180 d to 270 d); the improvement is not obvious. In the environment with high HA content (HA reagent content is 30%), the inhibitory effect of HA on the early stage of the hydration reaction (soaking time is less than 180 d) is strong, and the sample $q_u$ growth is slow. After the hydration reaction fully reacted with HA (soaked for about 180 d), the hydration reaction of the HA-containing cemented soil broke through the inhibition of HA to a certain extent, causing the HA-containing cemented soil $q_u$ begin to increase significantly.

UFC (the UFC particles in this study were all smaller than 32 μm) can promote the hydration reaction, and the addition of UFC makes the cemented soil $q_u$ significantly higher than that without UFC. According to related studies [30,33], cement particles with a particle size greater than 60 μm participate in the hydration reaction and produce limited hydration products. An increase in UFC proportion in the composite cement curing agent will generate more hydration products. The increase in UFC proportion in the composite cement curing agent will generate more hydration products, so the reinforcement effect of the composite cement curing agent on cemented soil is always stronger than that of OPC. In summary, the increase in UFC proportion improves the cement performance of the composite cement curing agent, thereby weakening the influence of HA on the cement curing effect. When the cemented soil strength is reduced by alkaline corrosion and disintegration, the cemented soil $q_u$ mixed with UFC is still higher than that of cemented soil without UFC, and the anti-destructive effect increases with the increase in the UFC proportion. The cemented soil sample can resist alkaline corrosion and disintegration with the addition of UFC.

Figure 8 shows the growth rate of samples $q_u$ with different HA reagent content when the UFC proportion increases from 0% to 10%. In the development stage of cemented soil strength (soaking time is 28 d, 90 d, 180 d, and 270 d), the addition of UFC causes the cemented soil $q_u$ with 30% HA reagent content to increase significantly. The reinforcement effect of the composite cement curing agent composed of UFC and OPC is higher than that of OPC, and the effect of HA on the cemented soil $q_u$ gradually weakens as the UFC proportion increases.

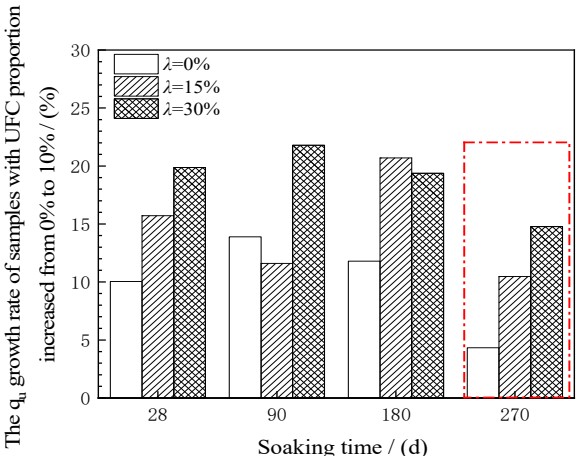

**Figure 8.** The UFC proportion increased from 0% to 10% $q_u$ growth rate of cemented soil.

Table 3 shows the growth rate data of cemented soil $q_u$ when the UFC proportion increases by 10%. $UFC_{0-10\%}$ is the growth rate of cemented soil $q_u$ when the UFC proportion increases from 0% to 10%. $UFC_{10-20\%}$ is the growth rate of cemented soil $q_u$ when the UFC proportion increases from 10% to 20%. $UFC_{20-30\%}$ is the growth rate of cemented soil $q_u$ when the UFC proportion increases from 20% to 30%. $UFC_{30-40\%}$ is the growth rate of cemented soil $q_u$ when the UFC proportion increases from 30% to 40%. $UFC_{40-50\%}$ is the growth rate of cemented soil $q_u$ when the UFC proportion increases from 40% to 50%. The cemented soil $q_u$ increases the most when the UFC proportion increases from 0% to 10% ($UFC_{0-10\%}$). Taking the maximum $q_u$ (270 d) in the strength development stage as the standard value, the proportions of samples $q_u$ with different UFC to samples without UFC is the strength improvement rate. The strength improvement rate is about the reduction rate of cement consumption, and the data analysis results are as follows: (1) cement consumption can be reduced by 4.3% to 12.9% when UFC proportion for 10% to 50% without HA; and (2) cement consumption is reduced 10.5% to 33.2% when the UFC proportion is 10% to 50% in the HA environment. UFC has a better reinforcement effect in peat soil environments, making the reduction in cement consumption more obvious.

**Table 3.** $q_u$ growth rate of cemented soil.

| Numbering | HA Reagent Content $\lambda$ of Soaking Time 28 d Samples/% | | | HA Reagent Content $\lambda$ of Soaking Time 90 d Samples/% | | | HA Reagent Content $\lambda$ of Soaking Time 180 d Samples/% | | | HA Reagent Content $\lambda$ of Soaking Time 365 d Samples/% | | |
|---|---|---|---|---|---|---|---|---|---|---|---|---|
| | 0 | 15 | 30 | 0 | 15 | 30 | 0 | 15 | 30 | 0 | 15 | 30 |
| $UFC_{0-10\%}$ | 10.0 | 15.7 | 19.9 | 13.9 | 11.6 | 21.8 | 11.8 | 20.7 | 19.4 | 4.3 | 10.5 | 14.8 |
| $UFC_{10-20\%}$ | 1.7 | 6.3 | 2.8 | 1.7 | 2.8 | 9.2 | 6.1 | 2.7 | 4.8 | 0.6 | 7.5 | 0.4 |
| $UFC_{20-30\%}$ | 0.8 | 3.1 | 6.5 | 5.4 | 1.0 | 0.5 | 2.8 | 4.8 | 6.7 | 1.3 | 1.4 | 5.4 |
| $UFC_{30-40\%}$ | 3.1 | 3.0 | 2.3 | 4.1 | 6.7 | 7.4 | 1.6 | 4.8 | 3.9 | 4.1 | 0.1 | 7.3 |
| $UFC_{40-50\%}$ | 0.3 | 0.4 | 1.3 | 2.3 | 1.3 | 2.8 | 3.6 | 2.4 | 2.4 | 2.1 | 6.6 | 2.3 |

The composite cement curing agent composed of UFC and OPC has a more substantial effect on peat soil foundation than OPC. The composite cement curing agent with a UFC proportion of 10% is economical and practical. This paper is the result of an indoor unconfined compressive strength test (UCS), which showed that the reinforcement effect of UFC in an HA environment is stronger than that of OPC. The application of actual engineering needs to combine indoor and field tests to obtain the optimal cement consumption and UFC proportion to reduce carbon emissions and the impact of the cement industry on the environment.

## 4. SEM and PCAS Analysis

Figure 9 shows the processing flow of SEM images of HA-containing cement soil with different ultra-fine cement (UFC) proportions using PCAS software. Researchers can use PCAS software to perform pore division and related data statistics on microscopic images [47,48]. The PCAS software binarized the SEM image, segmented the pores, and distinguished them with different colors. In the PCAS pore segmentation image, black represents cemented soil, and other colors represent pores. SEM images show that, as the UFC proportion increased, the permeability of pores in the sample decreased, the number of macropores gradually decreased, and the pores were filled from fibrous hydration products to cement.

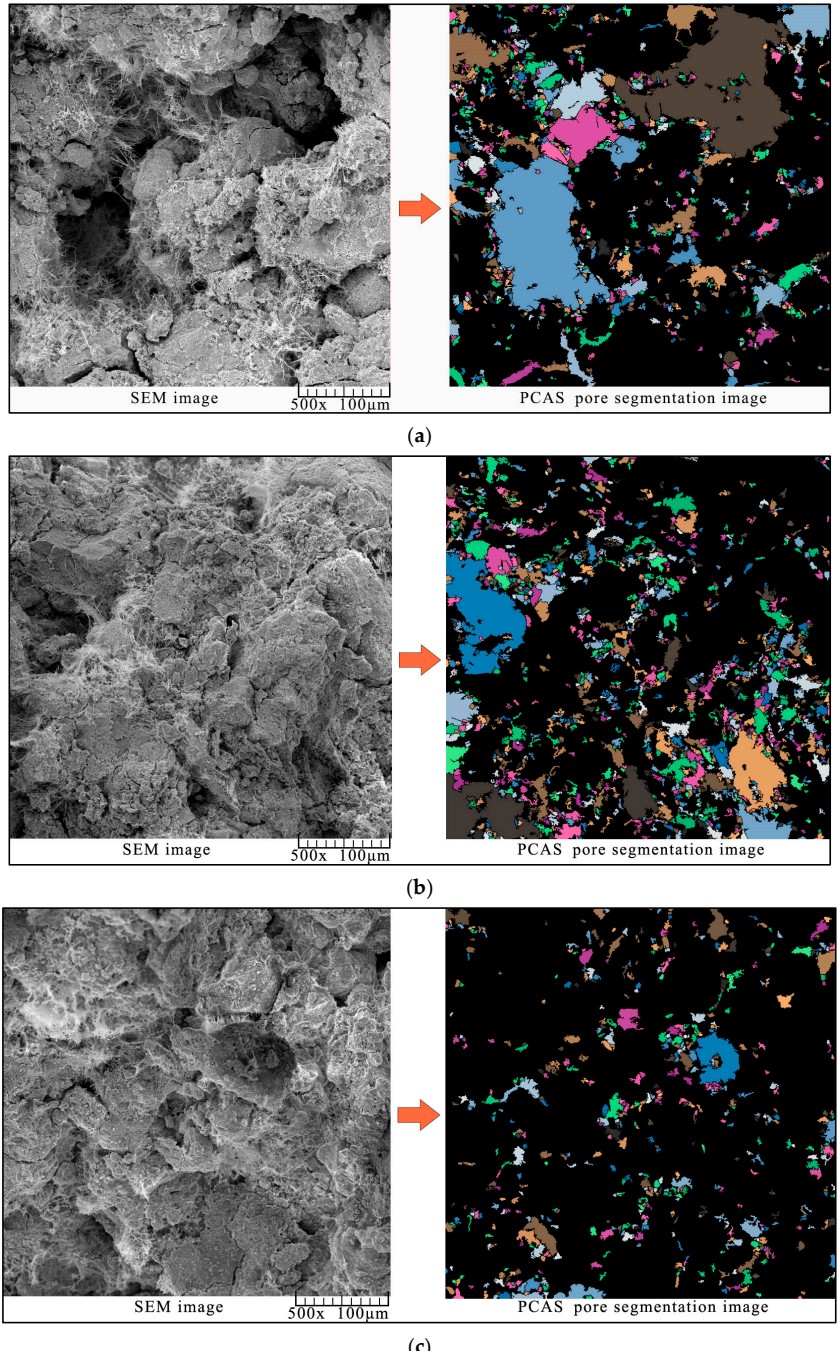

**Figure 9.** *Cont.*

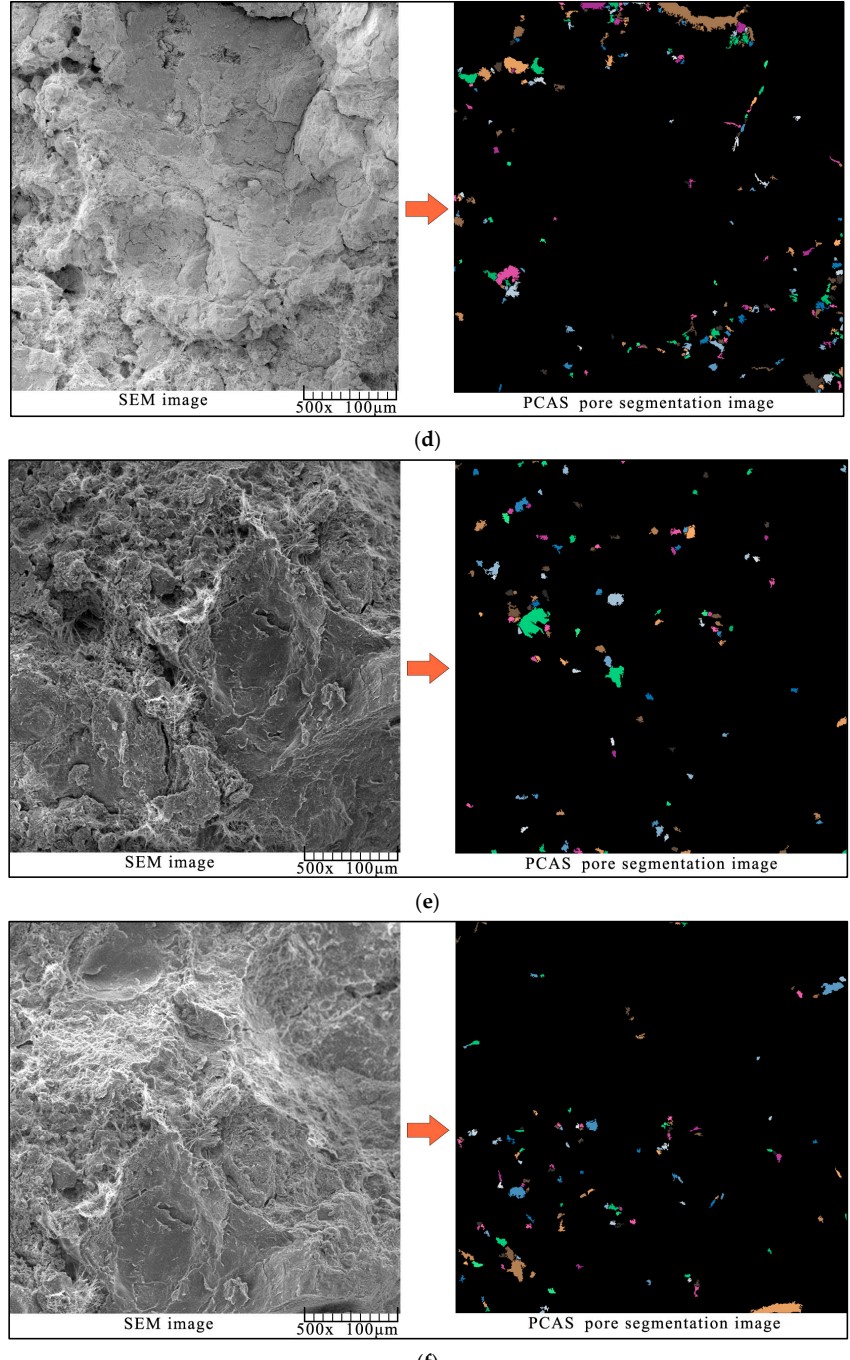

**Figure 9.** SEM images and PCAS pore segmentation images of HA-containing cemented soil with different UFC proportions $\gamma$. (**a**) $\gamma = 0\%$. (**b**) $\gamma = 10\%$. (**c**) $\gamma = 20\%$. (**d**) $\gamma = 30\%$. (**e**) $\gamma = 40\%$. (**f**) $\gamma = 50\%$.

The microstructure of HA-containing cemented soil without UFC presents obvious agglomerate connections. A network of hydration products or face-to-face contact connects the agglomerate. There are macropores between the agglomerate, so the network of hydration products cannot fill the pores. The microstructure of the HA-containing cemented soil samples changed significantly with the increase in the UFC proportion. The linking effect of agglomerates was enhanced and gradually formed larger structural elements.

Figure 10 shows the percentage of macropores in the total pore area of samples with different UFC proportions. The PCAS software divides the pores of the SEM image and obtains the characteristic pore data of the HA-containing cemented soil. In this paper, for the

convenience of analysis, according to the pore characteristics of HA-containing cemented soil, this paper defines pores with a pore diameter greater than 10 μm as macropores [43,49]. The test results show that increasing the UFC proportion can improve the microstructure of cemented soil. The macropore proportion in the sample decreased gradually with the increase in the UFC proportion. Macropore proportions decreased the most when UFC proportion increased from 0% to 10%. The reduction rate of the macropores proportion gradually decreased with the increase in the UFC proportion.

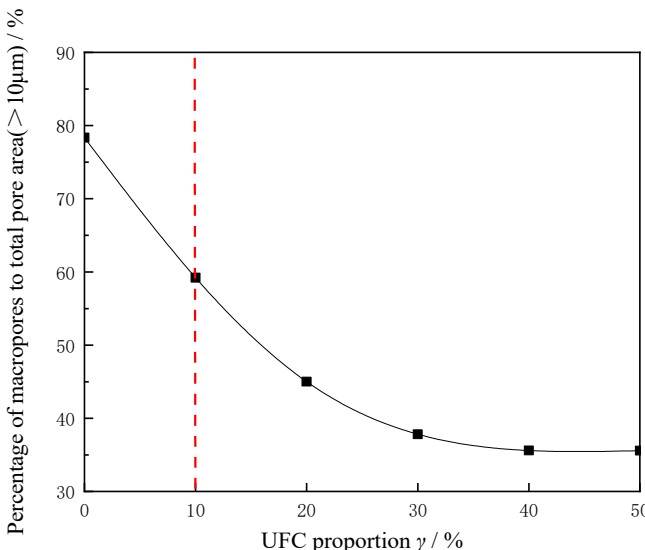

**Figure 10.** The percentage of macropores in the total pore area of samples with different UFC proportions.

The cementation effect of hydration products in HA-containing cemented soil gradually increases with the UFC proportion. The hydration products continue to cement aggregates to form structural elements and fill macropores, and the macropore proportion gradually decreases, increasing the HA-containing cemented soil's $q_u$. SEM results verified that UFC can enhance cemented soil's structure in an HA environment, and the cemented soil microstructure becomes compact. Therefore, adding UFC reduces the permeability and macropores proportion in HA-containing cemented soil. The proportion of macropores in HA-containing cemented soils strongly relates to the $q_u$. The unconfined compressive test (UCS), SEM test, and PCAS microscopic quantitative test technology show that the microstructure and strength of HA-containing cemented soil are improved when the UFC proportion is 10%. When the macropore proportion exceeds 10% in UFC, the reduction range becomes smaller, and the increase in strength also slows.

## 5. Conclusions

Different proportions of ultra-fine cement were added to ordinary Portland cement to reveal the effect of ultra-fine cement (UFC) on humic acid-containing cemented soil. Peat soils in different humic acid environments were simulated with humic acid reagents and cohesive soils, reinforced with composite cement curing agents with different UFC proportions. UFC's strengthening mechanism analysis was studied through an unconfined compressive strength test (UCS), scanning electron microscope test (SEM), and PCAS.

1.  HA will significantly reduce cemented soil's unconfined compressive strength ($q_u$), and HA will affect cemented soil's hydration reaction and strength development.
2.  The increase in the UFC proportion caused the cemented soil $q_u$ to gradually increase. Compared with OPC, the composite cement curing agent composed of UFC and ordinary Portland cement (OPC) had a better reinforcement effect. The improvement of cement performance by UFC is mainly reflected in accelerating the hydration

reaction and generating more hydration products. The cemented soil $q_u$ increased slowly with the increase in UFC proportion after UFC proportion exceeds 10%.

3. The $q_u$ increased gradually with the soaking time, reaching the highest strength at 270 d, and then decreased. UFC has a better effect on cemented soil reinforcement in environments with higher HA content than in non-HA environments. It is practical and economical to strengthen the peat soil with a composite cement curing agent with a UFC proportion of 10%.

4. The results of the SEM and PCAS microscopic quantitative test technology showed that UFC can reduce the macropore proportion and the connectivity of the pores. The agglomerates in HA-containing cemented soil gradually become larger structural elements with the increase in UFC proportion, which increases the strength of the sample.

5. The strength and microscopic tests show that UFC can improve the HA-containing cemented soil $q_u$, which is significant for engineering practice. UFC can significantly reduce cement consumption in an environment with high HA content. UFC can reduce $CO_2$ emissions and protect the environment by meeting engineering requirements. Based on previous studies of the properties of UFC, this study reveals the mechanism of UFC's influence on the mechanics and microstructure of HA-containing cemented soil [13,50,51]. In actual engineering, the research in this paper can be cited, and UFC can be used to reduce $CO_2$ emissions and environmental impact.

**Author Contributions:** Conceptualization, J.C. and F.L.; methodology, J.C.; formal analysis, F.L., W.D., and Y.G.; resources, Z.S., J.L., and G.L.; data curation, F.L.; writing—original draft, F.L.; visualization, J.C.; project administration, Z.S.; funding acquisition, J.C. All authors have read and agreed to the published version of the manuscript.

**Funding:** This research was funded by National Natural Science Foundation of China (Yunnan Province), grant number 41967035.

**Institutional Review Board Statement:** Not applicable.

**Informed Consent Statement:** Not applicable.

**Data Availability Statement:** The data used to support the findings of this study are included in the article.

**Conflicts of Interest:** The authors declare no conflict of interest.

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
