# Peer review of "Effect of Ultra-Fine Cement on the Strength and Microstructure of Humic Acid Containing Cemented Soil"

_sustainability, doi:10.3390/su15075923_

Round 1

Reviewer 1 Report

Journal: Sustainability (ISSN 2071-1050)

Manuscript ID: sustainability-2266232

Review Report

The authors presented an article on, “Study on the Effect of UFC on the Strength of HA-containing Cemented soil”. I think the article is well organized and suitable for the "Sustainability" journal. But the article will be ready for publication after a major revision. Comments are listed below.

1.      The similarity rate is 31%. It should be reduced.

2.      What's new in this study? What has been done that differs from previous work? It should be explained.

3.      In Table 1, the unit of natural density should be g/cm3.

4.      In Figure 3, the particle size analysis graph is given. The method of particle size analysis should be explained.

5.      How were the test materials mixed? Mixed with the help of a device? It should be explained.

6.      According to which standards are the compressive strength test performed?

7.      On page 6, line 170, the sentence "With the increase of HA reagent content, HA particles gradually replace some soil particles to become the skeleton of cemented soil" is repeated twice. It should be corrected.

8.      On page 9, line 290, "When the strength of cemented soil is reduced by alkaline corrosion and disintegration, the qu of cemented soil mixed with UFC is still higher than that of cemented soil without UFC, and the anti-destructive effect increases with the increase of the proportion of UFC" was repeated twice. It should be corrected.

9.      It should be explained which color represents which components in the SEM images in Figure 9.

10.  In Figure 9, changes in the microstructure should be further explained.

11.  The article contains numerous typographic and language errors. It should be corrected.

12.  The article should be rearranged by taking into account the journal writing rules and citation rules.

*** Authors must consider them properly before submitting the revised manuscript. A point-by-point reply is required when the revised files are submitted.

Author Response

Many thanks to the reviewers for their suggestions on our manuscript. The revision comments were very professional and considerably improved our manuscript. We have carefully reviewed the comments and have revised the manuscript accordingly.

Point 1: The similarity rate is 31%. It should be reduced.

Response 1: Thank you very much for reminding us. We have partially revised the content of the manuscript based on the report. We have adjusted the parts in blue to make the article more integrated and logical.

Point 2: What's new in this study? What has been done that differs from previous work? It should be explained.

Response 2: Based on previous studies, the influence of ultra-fine cement (UFC) on the strength development of humic acid containing cemented soil is revealed, and the possibility of reducing the cement consumption in actual engineering is explored. This paper focuses on studying the strength development law of cement soil using UFC in a humic acid (HA) environment.

Make revisions in page 2, line 87-89 of the manuscript.

Point 3: In Table 1, the unit of natural density should be g/cm3.

Response 3: We have modified the units of Natural Density in Table 1.

Make revisions in page 3, line 108-109 of the manuscript.

Point 4: In Figure 3, the particle size analysis graph is given. The method of particle size analysis should be explained.

Response 4: We added a sentence describing the experimental method for particle size analysis.

Laser particle size analysis is performed on the test materials using a laser particle size analyzer (Malvern Mastersizer 2000).

Make revisions in page 4, line 116-117 of the manuscript.

Point 5: How were the test materials mixed? Mixed with the help of a device? It should be explained.

Response 5: The test material is put into the blender and stirred evenly, then distilled water is added to continue stirring.

Make revisions in page 4, line 144-146 of the manuscript.

Point 6: According to which standards are the compressive strength test performed?

Response 6: The unconfined compressive strength test (UCS) is carried out on the samples according to the " Standard for geotechnical testing method " (GB/T 50123-2019).

Make revisions in page 5, line 158-159 of the manuscript.

Point 7: On page 6, line 170, the sentence "With the increase of HA reagent content, HA particles gradually replace some soil particles to become the skeleton of cemented soil" is repeated twice. It should be corrected.

Response 7: We removed duplicate sentences.

Make revisions in page 6, line 177-178 of the manuscript.

Point 8: On page 9, line 290, "When the strength of cemented soil is reduced by alkaline corrosion and disintegration, the qu of cemented soil mixed with UFC is still higher than that of cemented soil without UFC, and the anti-destructive effect increases with the increase of the proportion of UFC" was repeated twice. It should be corrected.

Response 8: We removed duplicate sentences.

Make revisions in page 9, line 280-282 of the manuscript.

Point 9: It should be explained which color represents which components in the SEM images in Figure 9.

Response 9: In the PCAS pore segmentation image, black represents cemented soil, and other colors represent pores.

Make revisions in page 11, line 326-327 of the manuscript.

Point 10: In Figure 9, changes in the microstructure should be further explained.

Response 10: The microstructure of humic acid (HA) containing cemented soil without ultra-fine cement (UFC) presents obvious agglomerate connections. A network of hydration products or face-to-face contact connects the agglomerate. There are macropores between the agglomerate, so the network of hydration products cannot fill the pores. The microstructure of the HA-containing cemented soil samples changed significantly with the increase in the UFC proportion. The linking effect of agglomerates is enhanced and gradually forms larger structural elements.

Make revisions in page 11, line 330-335 of the manuscript.

Point 11: The article contains numerous typographic and language errors. It should be corrected.

Response 11: We typeset and grammar-checked the manuscript and made revisions.

Point 12: The article should be rearranged by taking into account the journal writing rules and citation rules.

Response 12: We revised the manuscript according to the journal's writing and citation rules.

Thank you again for taking the time out of your busy schedule to review the manuscript for us.

Reviewer 2 Report

  1. The paper titled “Study on the Effect of UFC on the Strength of HA-containing Cemented soil” described the reinforcement effect of peat soil foundation treated by cemented soil mixing method 9 is mainly affected by humic acid. The manuscript has potential but need major changes before consideration
  2. Title is not comprehensive as do not add any abbreviations in the title so modify accordingly
  3. Abstract is written poor as lack study background and clear methodology so elaborate study background, objective, results and outcomes with practical application. Also add the different stakeholders for which findings useful at the end of abstract with particular focus of findings application  
  4. Introduction is written good and has made the foundation for the paper however, add the regulatory requirements for the carbon emission and different soils type contribution along with impact on enevironment.
  5. Table 1 “Physical properties of soil” heading need elaborations
  6. Material and method section is written is good
  7. What is the criteria for selecting different soil samples that need to be elaborated?
  8. Statistical analysis is missing. Justify or add ?
  9. Results are written in a good way however add the latest references to support findings of the paper
  10. In conclusion major focus should be on findings with practical application
  11. Grammatical mistakes observed on few places so there is need to go through the paper for language and grammatical mistakes

Author Response

We sincerely thank the editor and all reviewers for their valuable feedback that we have used to improve the quality of our manuscript. We have carefully reviewed the comments and have revised the manuscript accordingly. The following are the corresponding revisions.

Point 1: The paper titled “Study on the Effect of UFC on the Strength of HA-containing Cemented soil” described the reinforcement effect of peat soil foundation treated by cemented soil mixing method is mainly affected by humic acid. The manuscript has potential but need major changes before consideration.

Response 1: Thanks for your valuable comments. We have revised the manuscript based on the comments.

Point 2: Title is not comprehensive as do not add any abbreviations in the title so modify accordingly.

Response 2: We have reconsidered the title of the manuscript and made revisions.

This paper mainly studies the mechanism of ultra-fine cement (UFC) on the strength development of simulated peat soil. Scanning electron microscopy (SEM) and PCAS microscopic quantitative analysis techniques complement related mechanisms. We wanted to highlight the study of "strength" in the title. We will reconsider if you still feel that the title needs to be more comprehensive.

Thank you so much for your suggestion.

Point 3: Abstract is written poor as lack study background and clear methodology so elaborate study background, objective, results and outcomes with practical application. Also add the different stakeholders for which findings useful at the end of abstract with particular focus of findings application.

Response 3: Thanks for the reminder. We rewrote the research background and methodology in the Abstract.In the Abstract last sentence, we highlighted the application range and future application prospects of ultrafine cement.

Make revisions in page 1, line 28-29 of the manuscript.

Point 4: Introduction is written good and has made the foundation for the paper however, add the regulatory requirements for the carbon emission and different soils type contribution along with impact on environment.

Response 4: We supplement the application fields and application prospects of ultra-fine cement (UFC) in the last paragraph of the Introduction.

Make revisions in page 2, line 92-96 of the manuscript.

Point 5: Table 1 “Physical properties of soil” heading need elaborations.

Response 5: Thanks for the reminder. We revised the title of Table 1.

Make revisions in page 3, line 108 of the manuscript.

Point 6: Material and method section is written is good.

Response 6: Thank you for your affirmation of these two parts.

Point 7: What is the criteria for selecting different soil samples that need to be elaborated?

Response 7: The humic acid (HA) content of peat soil in the Dianchi Lake area is about 2.36%~28.13%. The HA reagent content λ (0%, 15%, 30%) is set according to the actual HA content, and the HA reagent is mixed with cohesive soil to simulate different peat soil environments.

We have described in detail the setting of humic acid incorporation.

Make revisions in page 4, line 133-136 of the manuscript.

Point 8: Statistical analysis is missing. Justify or add?

Response 8: We have added statistics analysis.(Table 3)

Table 3 is used for Justify.

Table 3 justifies the practicability of 10% UFC in the composite cement curing agent.

The strength improvement rate is about the reduction rate of cement consumption, and the data analysis results are as follows: (1) Cement consumption can be reduced by 4.3% to 12.9% when ultra-fine cement (UFC) accounts for 10% to 50% without humic acid (HA); (2) Cement consumption is reduced 10.5% to 33.2% when the UFC proportion is 10% to 50% in the HA environment. UFC has a better reinforcement effect in peat soil environments, which makes the reduction of cement consumption more obvious.

Make revisions in page 10, line 304-309 of the manuscript.

Point 9: Results are written in a good way however add the latest references to support findings of the paper.

Response 9: We have added up-to-date references in the Conclusions (Results) section to support the paper's findings.

Make revisions in page 14, line 393-396 of the manuscript.

Point 10: In conclusion major focus should be on findings with practical application.

Response 10: We highlight findings for practical applications in the Abstract, Introduction, and Conclusion sections.

We have made adjustments to the Conclusions section.

Make revisions in page 1, line 28-29 of the manuscript.( Abstract)

Make revisions in page 2, line 92-96 of the manuscript. (Introduction)

Make revisions in page 10, line 304-309 of the manuscript.

Make revisions in page 14, line 393-396 of the manuscript. (Conclusion)

Point 11: Grammatical mistakes observed on few places so there is need to go through the paper for language and grammatical mistakes.

Response 11: We revised the sentences of the manuscript and invited students with study abroad-backgrounds to review it.

We thank you again for your review comments.

Round 2

Reviewer 1 Report

Journal: Sustainability (ISSN 2071-1050)

Manuscript ID: sustainability-2266232

 Review Report R2#

  The authors completed the requested corrections. Therefore, in my opinion, this article can be accepted for publication in the "Sustainability" journal in its final form.

Author Response

Point 1: The authors completed the requested corrections. Therefore, in my opinion, this article can be accepted for publication in the "Sustainability" journal in its final form.

Response 1: We are very grateful for the reviewer's endorsement. Based on your suggestions, this manuscript has been revised and improved. Your Comments and Suggestions is very professional, and all of us authors thank you for your work.

We have rewritten the blue part appropriately.The meaning of the blue part has stayed the same.

Reviewer 2 Report

All the changes are addresses except the UFC abbreviation in title that need to be corrected

Author Response

Point 1: All the changes are addresses except the UFC abbreviation in title that need to be corrected.

Response 1: Thank you for your Comments and Suggestions. We have made changes to the title. We recognize that using no abbreviations in the title would be better. We thank you again for your work.

We have rewritten the blue part appropriately. The meaning of the blue part has stayed the same.